# MicroRNAs as Biomarkers in Spinal Muscular Atrophy

**DOI:** 10.3390/biomedicines12112428

**Published:** 2024-10-23

**Authors:** Maruša Barbo, Damjan Glavač, Gregor Jezernik, Metka Ravnik-Glavač

**Affiliations:** 1Institute of Biochemistry and Molecular Genetics, Faculty of Medicine, University of Ljubljana, SI-1000 Ljubljana, Slovenia; marusa.barbo@mf.uni-lj.si; 2Center for Human Genetics & Pharmacogenomics, Faculty of Medicine, University of Maribor, SI-2000 Maribor, Slovenia; damjan.glavac@mf.uni-lj.si (D.G.); gregor.jezernik1@um.si (G.J.); 3Department of Molecular Genetics, Institute of Pathology, Faculty of Medicine, University of Ljubljana, SI-1000 Ljubljana, Slovenia

**Keywords:** spinal muscular atrophy, SMA, circulating miRNA, biomarkers, molecular networks

## Abstract

Spinal muscular atrophy (SMA) is a severe neurodegenerative disease caused by the loss of the survival motor neuron (SMN) protein, leading to degeneration of anterior motor neurons and resulting in progressive muscle weakness and atrophy. Given that SMA has a single, well-defined genetic cause, gene-targeted therapies have been developed, aiming to increase SMN production in SMA patients. The SMN protein is likely involved in the synthesis of microRNAs (miRNAs), and dysregulated miRNA expression is increasingly associated with the pathophysiology of SMA. Currently, there is a lack of reliable biomarkers to monitor SMA; therefore, the search for novel SMA biomarkers, including miRNAs, is crucial as reliable tools are needed to track disease progression, predict the response to therapy and understand the different clinical outcomes of available treatments. In this review, we compile data on miRNAs associated with SMA pathogenesis and their potential use as biomarkers. Based on current knowledge, the most frequently deregulated miRNAs between SMA patients and controls, as well as pre- and post-treatment in SMA patients, include miR-1-3p, miR-133a-3p, miR-133b, and miR-206. These findings offer promising possibilities for improving patient classification and monitoring disease progression and response to treatment. Additionally, these findings provide insights into the broader molecular mechanisms and networks of SMA that could inform the development of future therapeutic strategies.

## 1. Introduction to Spinal Muscular Atrophy

Spinal muscular atrophy (SMA) is a severe neurodegenerative disease (ND) characterized by anterior motor neuron degeneration, which results in increasing muscular weakening and paralysis [1]. SMA is inherited in an autosomal recessive manner [2] and is the most common hereditary cause of neonatal death [3], with an incidence of ∼1 in 10,000 live births [4].

### 1.1. Genetic Basis

SMA is mostly caused by homozygous deletion of exons 7 and 8, or, rarely, only of exon 7, within the survival motor neuron 1 (*SMN1*) gene, which results in survival motor neuron (SMN) protein deficiency [5,6]. Other *SMN1* mutations can be detected in certain cases, usually with an *SMN1* deletion on the other allele [7]. Despite the loss of *SMN1*, SMA patients maintain varied numbers of copies of the paralogous gene, referred to as *SMN2*. *SMN2*, however, is unable to compensate for the loss of function of full-length *SMN1* [5,8]. The *SMN1* and *SMN2* genes differ by 14 nucleotides and one insertion, where the most important nucleotide change is a C-to-T transition located in the coding region, in exon 7 [9]. Because of the nucleotide substitution, exon 7 is excluded in approximately 90% of *SMN2* transcripts, which results in shortened unstable protein (SMNΔ7) that is rapidly destroyed (Figure 1) [10]. In SMA patients, the number of *SMN2* copies ranges from 0 to 5 [11]. A lower number of *SMN2* copies is associated with lower levels of full-length SMN protein, while the complete absence of the SMN protein is lethal [12]. Therefore, the severity of SMA varies, at least partially, depending on the number of *SMN2* copies that patients carry [13]. Other factors impacting SMA’s severity and symptoms involve positive or negative disease modifiers, such as rare *SMN2* variants (e.g., c.859G  >  C) and independent factors such as plastin 3 and neurocalcin delta [14].

While ubiquitously expressed in the cell nucleus and cytoplasm, the SMN protein serves a critical role in cell homeostasis and takes part in a variety of cellular mechanisms, including spliceosomal machinery assembly, protein translation, RNA metabolism, cytoskeletal dynamics, endocytosis and autophagy [15,16]. In particular, the biogenesis of ribonucleoproteins (RNPs) is considered one of the main roles of the SMN [17].

The motor neurons of the spinal cord continue to express high levels of the SMN protein, not only during the gestational and neonatal period but also throughout life. Therefore, they are extremely susceptible to SMN depletion as in SMA disease [6,18]. It is believed that SMN deficiency simultaneously disturbs a number of key neuronal processes, i.e., neuron development, growth, and survival [16].

Furthermore, due to the widespread expression of SMN and its various functions, loss of SMN can result in systemic disease that extends beyond the motor neuron [15]. The acceptance of SMA as a multi-organ disease is a consequence of the finding of various impairments of the heart, blood vessels, liver, pancreas, gastrointestinal system, thymus, spleen and bones ([19,20] and references therein).

### 1.2. Classification and Clinical Manifestation

SMA has been traditionally classified by the International Spinal Muscular Atrophy Consortium into three main types (types 1–3) based on the age of onset and maximal motor milestone accomplished [14,21,22]. Some experts, however, advocate for an expanded categorization that encompasses additional phenotypes—including congenital (type 0) [23] and adult-onset SMA (type 4) (Table 1) [24].

The original classification of the SMA types refers only to the best motor status of patients and not to their current functional status. Therefore, in the functional classification, SMA patients are divided into three functional levels according to their current neurological status: non-sitter, sitter, and walker. Considering that type 3 patients who have lost their ability to walk have many aspects in common with type 2 patients, they can be grouped as “sitters”. Type 3 patients who are still able to walk are referred to as “walkers” and type 1 patients are referred to as “non-sitters”. This classification emphasizes the current functional level and the therapeutic response while acknowledging the SMA phenotype as a continuum [25,30,31].

### 1.3. Disease Management

Since SMA is an ND with a single genetic cause, gene-targeted therapies have been developed for SMA treatment. There are currently three therapies approved by the U.S. Food and Drug Administration (FDA) and the European Medicines Agency (EMA), which are designed to increase SMN production. Nusinersen (Spinraza; Biogen), as the first approved SMA treatment, is an antisense oligonucleotide designed to modify the splicing of *SMN2* pre-mRNA [32,33]. Administered by intrathecal injection, as it cannot cross the blood–brain barrier, it significantly increases exon 7 inclusion and is indicated for use in pediatric and adult patients [34,35]. Onasemnogene abeparvovec (Zolgensma; Novartis) is a gene replacement therapy where the viral vector is used to deliver the exogenous gene. It is approved in the United States for intravenous administration to SMA patients under two years of age and in type 1 SMA patients with up to 3 *SMN2* copies in the European Union [36,37]. Risdiplam (Evrysdi; Roche) functions as an *SMN2*-splicing modifier, which binds directly to *SMN2* pre-mRNA and promotes the inclusion of exon 7 and therefore full-length SMN protein synthesis [38,39]. Administered orally as a liquid formulation, it was approved by the FDA in 2020 for the treatment of SMA patients aged over two months and by the EMA in 2021 for patients with type 1, 2, or 3 SMA or 1–4 *SMN2* copies at two months of age [38,39].

Despite the significant progress that the SMA therapies, such as nusinersen, have made in the treatment of SMA disease [40,41], ongoing research in the field remains essential. As a functional SMN protein is crucial, especially during the early developmental stages [42], current treatments may not be sufficient to fully restore neuromuscular function in older patients and patients in the later stages of the disease. Thus, the long-term neuro-muscular health of SMA patients largely depends on novel biomarkers, which would enable prediction of the treatment response and effective tracking of disease progression.

Given the link between the components of the SMN complex and microRNA (miRNA), as well as the increasingly recognized role of miRNAs in the pathophysiology of SMA [43,44], this review aims to provide insights into the current state of research and bridge the gap due to the absence of biomarkers, which are needed to improve patient classification, track SMA progression, and monitor the response to treatment. We aim to address the gap in the literature, as similar narrative [45] and systematic reviews [46,47,48] have not yet focused specifically on the role of miRNA biomarkers in SMA. We are particularly interested in studies that compared miRNA discovery profiling in blood samples from SMA patients with those from healthy controls. These investigations were able to find potential diagnostic or susceptibility biomarkers for SMA based on circulating miRNAs. Furthermore, we explore studies on the miRNA signatures in nusinersen-treated SMA patients, aiming to determine the effects of therapeutic intervention on the miRNA profile by linking molecular data to longitudinal clinical responses to therapy, as reflected by changes in motor function scores.

## 2. SMA Biomarkers

The expansion of treatment options has created a corresponding need for biomarkers that can guide therapy and enhance disease monitoring. Significant efforts have been made to identify suitable markers, and many potential biomarkers have been discovered for diagnostic, prognostic, and predictive purposes. Several review studies attempted to evaluate the most appropriate biomarkers suitable for clinical application [45,46,47,48,49].

The main findings of the systematic review on the clinical utility of blood biomarkers in patients with SMA by Navarrete-Opazo et al. [46] were that SMN-related biomarkers exhibited significant variability between patients and cell types, with considerable overlap between SMA phenotypes and healthy controls, so there is not yet sufficient evidence to confirm the clinical utility of SMN-related biomarkers for predicting the disease severity in SMA. On the other hand, DNA methylation analysis effectively differentiated between patients with mild to moderate SMA and healthy controls. The levels of plasma phosphorylated neurofilament heavy chain (pNfH) increased with the disease severity but showed a significant decline following nusinersen treatment. Thus, pNfH appears to be a promising indicator of disease activity and treatment response in SMA.

The systematic review by Gavriilaki et al. [47] of real-world observational studies suggests that several biomarkers, including serum creatinine, creatine kinase activity, as well as cerebrospinal fluid (CSF) Aβ42, glial fibrillary acidic protein (GFAP) concentrations, ulnar compound muscle action potential (CMAP), and single motor unit potential (SMUP) amplitude, may help predict the therapeutic response in patients with SMA types 2, 3, and 4 who are over the age of 11 and undergoing treatment with nusinersen. In a review and meta-analysis, Gavriilaki et al. [48] summarized the existing data on the long-term progression of motor function tests, muscle strength, lung function, neurophysiological measurements, serum markers, and imaging biomarkers in adolescents and adults with SMA. The evidence, albeit of moderate quality, suggests that lung function tests and serum biomarkers are inadequate for capturing the natural progression of the disease in this group. For other biomarkers, the presence of conflicting results and the variability among studies hindered any definitive conclusions about their effectiveness as indicators of disease activity. Nevertheless, the pooled analysis indicated that the CMAP and motor unit number index (MUNIX) scores in SMA patients with longer disease durations were significantly different from those of healthy controls. Lapp et al. [45] reviewed the candidate biomarkers, including SMN-related markers that show promise as diagnostic biomarkers, neurodegeneration and imaging markers as prognostic biomarkers, electrophysiological markers as predictive biomarkers, and muscle integrity markers as response markers, particularly in the context of upcoming muscle-targeting therapies. However, no single measure seems to be suitable for covering all the biomarker categories and pathologies involved in SMA.

The evaluation of SMA biomarkers in the above-mentioned reviews relied primarily on the clinical outcomes, including motor function, electrophysiological tests, and respiratory function, as well as some molecular markers, including quantification of the SMN mRNA or protein levels, plasma pNfH, and serum creatinine levels.

### 2.1. MicroRNAs in SMA

Researchers have discovered that many SMN-associated modifiers play a role in both coding and non-coding RNA (ncRNA) processing, so studies of the molecular mechanisms underlying SMA are now focusing on RNA metabolism. MiRNAs are a subclass of short regulatory ncRNAs with a length of about 22 bp and are the most studied in terms of their effect on post-transcriptional gene expression [50]. By coupling with their target mRNAs, they control the expression of several target genes at the same time and contribute to the activity of the associated signaling pathways. The findings suggest that they are essential for the development, function, and survival of spinal motor neurons, particularly the cytoskeletal structure, synapse formation, and axonal growth [49,51,52,53,54].

Dysregulation or abnormal expression of miRNAs has been shown to play an important role in the pathophysiology of various NDs [55,56], including SMA [53,57]. To elaborate, alteration of motor neuron-specific miRNA expression is one mechanism that appears to be involved in the selective vulnerability of motor neurons, as identified in SMA models [53]. The SMN protein, which serves an important function in RNA processing, is likely to be involved in the biogenesis of miRNA, as it binds directly to proteins involved in miRNA production and function, including fragile X mental retardation protein (FMRP), KH-type splicing regulatory protein (KSRP), and fused in sarcoma/translocated in liposarcoma (FUS/TLS) [58,59,60]. Therefore, SMN deficiency affects miRNAs and their target mRNAs in motor neurons [2]. Since one miRNA can affect several genes at the same time, this may also be a plausible explanation for why SMN depletion affects so many pathways [53].

Recently, miRNAs have not only been increasingly associated with the pathogenesis of SMA but have also been investigated as potential biomarkers for monitoring SMA progression or treatment response, as well as promising therapeutic targets for SMA ([44,53] reviewed in [49]). MiRNAs have attracted significant interest as accessible circulating biomarkers, as they can be detected in various biofluids—such as CSF, serum, plasma, saliva, and urine—through non-invasive methods. They are now being studied as innovative clinical biomarkers for the prognosis of several diseases, including various cancers, cardiovascular diseases, and neurodegenerative disorders [61,62,63].

### 2.2. Methods for miRNA Detection

Due to their short sequence length, high degree of homology and low copy number, miRNAs are not only difficult to detect but also require highly specific methods for their selective quantification [64]. Conventional techniques for the detection of miRNAs include northern blot, reverse transcription quantitative polymerase chain reaction (RT-qPCR), microarray and next-generation sequencing (NGS).

The RT-qPCR method is a gold standard for the detection of miRNAs. It offers high sensitivity and single nucleotide specificity while being cost-effective at the same time; however, it enables only relative quantification [65,66,67]. The method involves two steps: RT to synthesize cDNA from the miRNA target, followed by PCR amplification monitored by fluorescence. However, the adaptation of RT-qPCR for miRNA quantification requires special primers, such as stem-loop primers [66] or two-tailed primers [68], as well as optimizations such as poly(A) strategies [65] or ligation-based PCR methods [69,70].

Another technique for miRNA detection is next-generation sequencing (NGS), a powerful tool for miRNA profiling. It enables the discovery of numerous miRNAs by sequencing millions to billions of RNA sequences within a short time frame [71]. The method includes RNA extraction, adapter ligation, RT, PCR amplification and sequencing. One of its major advantages is its high multiplexing capability, which enables the detection of all the RNAs in a sample without specific probes or primers [72]. However, NGS has some limitations, including the need for pre-amplification, potential sequence dependent biases, lower sensitivity for rare miRNAs and high cost. Nevertheless, it is emerging as a benchmark for identifying disease-related miRNA signature, as shown in the following studies in the field of SMA [73,74,75,76]

## 3. Circulating miRNAs as Potential Biomarkers for SMA

Circulating miRNAs are currently emerging as novel clinical indicators for predicting the outcome of several diseases [61,62,77]. MiRNAs can be secreted from cells either actively or as a result of external stimuli or damage, leading to the fluctuation of their levels in biofluids that are nevertheless relatively stable. This stability makes miRNAs a promising option as biomarkers and therapeutic targets. In addition to their role in spinal motor neuron function and survival, they have also been found in muscles and neuromuscular junctions. Moreover, the detected changes in the levels of several tissue-specific miRNAs may prove useful for monitoring the SMA progression or therapeutic response. In fact, because of the SMN’s role in regulating miRNA expression, miRNAs have emerged as a viable biomarker for SMA that can be easily detected in diverse human biofluids using non-invasive approaches [49].

### 3.1. Circulating miRNAs as Potential Diagnostic Biomarkers for SMA

Catapano et al. evaluated three selected miRNAs—miR-9, miR-206, and miR-132—in type 2 and type 3 SMA patients aged 4 to 14 years. They found upregulated expression of miR-9 and miR-132 in serum from patients with type 2 SMA compared with healthy controls. However, these miRNAs were found at an intermediate level in the serum of patients with type 3 SMA, with no statistically significant difference. The serum miR-206 levels in SMA patients were not significantly different from those in the control group. Similarly, the authors discovered no significant relationship between the miRNA levels studied and the motor functional ability of the patients, as assessed by the Hammersmith Functional Motor Scale (HFMS) [78].

Malacarne et al. investigated the expression of selected muscle-specific miRNAs (myomiRs)—miR-206, miR-133a, miR-133b, and miR-1—in the serum of pediatric SMA patients, aged 6.86 ± 3.33 years, to establish the role of myomiRs as noninvasive biomarkers. Their findings revealed dysregulation of myomiRs in patients’ serum, namely overexpression of miR-206 in human serum [79]. The same research group previously reported a decrease in these myomiRs during the course of the disease with nusinersen treatment in the serum of infantile SMA patients, which could be due to an improvement in muscle condition and performance [80].

Abiusi and co-workers first analyzed the miRNome of muscle samples from SMA patients (median age of 1.8 years) and compared it with controls, followed by evaluation of over 100 miRNAs, which were found to be differentially expressed in muscle samples. They identified 24 differentially expressed miRNAs and validated them in a larger cohort of 51 SMA patients (mean age of 17.3 ± 19.2 years). The results revealed three miRNAs—miR-181a-5p, miR-324-5p, and miR-451a—that were significantly overexpressed in the serum of SMA patients. The authors believe that these miRNAs are actively released by the skeletal muscle; however, they have not previously been described in SMA. Compared with the results previously published by Catapano et al., miR-206 was found to not be differentially expressed, miR-9 was virtually undetectable in the cohort’s serum samples, whereas miRNA-132 was not tested in this study [73].

### 3.2. Circulating miRNAs as Potential Prognostic and Therapeutic Biomarkers for SMA

Despite the advancement of innovative molecular and gene therapies for SMA, nusinersen continues to be the most extensively studied disease-modifying SMA treatment [81]. The drug has been shown to improve the clinical outcomes in young type 1 SMA patients [82,83]; however, the response in people with later-onset SMA (types 2 and 3) is varied among individuals. Some patients respond well, others only partially, with 30–40% showing a clinically significant improvement [40,75,84]. To further explore the utility of circulating miRNAs as biomarkers for SMA, studies have been conducted to explore the potential of miRNAs as prognostic biomarkers and to better understand the variability of the response to nusinersen treatment [75].

A study by Bonanno et al. looked at using certain serum-derived myomiRs, namely miR-133a, miR-133b, miR-1, and miR-206, as biomarkers to monitor the effects of nusinersen therapy in pediatric SMA type 2 and 3 patients with a mean age of 5.18 ± 3.42 years. The results showed a significant decrease in miR-133a, miR-133b, and miR-1 after 6 months of treatment. Additionally, a trend toward a decrease in miR-206 was observed compared to pre-nusinersen levels; however, the change was not statistically significant. This aligns with the findings of Catapano et al., who reported no significant differences in serum-derived miR-206 expression between SMA patients and healthy individuals [78]. Notably, there was a substantial positive association between these myomiRs, suggesting synergistic expression in response to therapy. Moreover, the reduction in the miR-133a levels was observed to correlate with disease improvement, as evaluated by the Hammersmith Functional Motor Scale Expanded (HFMSE), indicating that miR-133a is a potential biomarker for therapeutic effect. The authors believe that the observed reduction in the myomiR levels in SMA patients treated with nusinersen shows the drug’s effect on muscle homeostasis, which may be trackable by the serum myomiR levels [80].

Welby et al. performed a study to investigate whether miRNAs in CSF from SMA patients could be used as markers of neuronal and glial health after nusinersen therapy. The authors used a filtering approach to focus the transcriptomic profiling on miRNAs associated with SMA or relevant to motor neuron degeneration, including miR-132, miR-218, miR-9, miR-23a, miR-431, and miR-146a. The study included a total of 12 SMA patients, ranging in age from 2 months to 20 years, with SMA type 1 through type 3. They observed an increase in miRNA expression in the nusinersen-treated CSF samples from the patients, which was associated with improved motor performance in the SMA type 1 and 2 patients. Moreover, miR-146a produced by astrocytes, which is known to be toxic to SMA motor neurons [85], remained upregulated in the CSF samples from the nusinersen-treated patients [86].

In the following study by Magen and coworkers [74], results were found that were consistent with the data of Bonanno et al. [80], suggesting that serum myomiR expression could serve as a predictor of the treatment response to nusinersen. First, their results showed the presence of myomiRs in the CSF. Second, the CSF myomiR levels were lower in SMA patients who responded positively to treatment with nusinersen than in those who did not. Their study included 34 individuals with late-onset SMA, with a median age of 11.0 year (range: 1.7–56.6 years). The study employed unbiased NGS to identify miRNAs in CSF as potential biomarkers for assessing the response to nusinersen. In summary, their data revealed lower baseline CSF levels of miR-133a-3p and miR-206, either alone or in combination, predicting a more positive clinical outcome for nusinersen 6 months after treatment initiation. Furthermore, the miR-206 levels were shown to be inversely associated with the HFMSE score. These results may aid in determining which patients are more likely to respond to nusinersen treatment [74].

In a similar study by Zaharieva et al., NGS-based miRNA profiling identified 69 miRNAs that were significantly dysregulated in SMA type 2 and type 3 patients, aged 4.7 to 16.1 years (median: 10.2 years), compared with healthy controls. The number of miRNAs discovered in SMA patients was lower than in controls. The dysregulated miRNAs were subsequently investigated in a therapeutic response cohort. The cohort consisted of SMA type 1 patients with a median age of 19 months, who were treated with nusinersen at three different points in time. Certain miRNAs (miR-107, miR-142-5p, miR-335-5p, miR-423-3p, miR-660-5p, miR-378a-3p, and miR-23a-3p) were shown to be linked to the response to nusinersen administration. In the therapeutic response cohort, the baseline levels of seven miRNAs were associated with the response to nusinersen treatment at 2 and 6 months; moreover, the levels of the first six miRNAs at 2 months of treatment were linked to the response at 6 months of treatment. Thus, these miRNAs might be useful for predicting and tracking nusinersen therapy responses. Additionally, the miRNA levels studied were linked to SMA clinical outcome indicators, for instance, the CHOP-INTEND (Children’s Hospital of Philadelphia Infant Test of Neuromuscular Disorders) and HINE (Hammersmith Infant Neurological Examination Section 2) scales. Both the CHOP-INTED and HINE scales were associated with miR-378a-3p, and furthermore, increased miR-378a-3p levels were related to improved motor function in individuals treated with nusinersen. Analysis of seven miRNAs showed that they target genes involved in the neurotrophin, mTOR, and FoxO signaling pathways, which have been associated with the regulation of cell survival, apoptosis, and autophagy [75].

In the exploratory study by D’Silva et al., miRNA sequencing was used to examine the differential expression of miRNAs in the CSF of six children with SMA, aged 16 to 390 days at the collection of first CSF specimen, who were treated with nusinersen. The results demonstrated that 14 miRNAs were significantly altered in the CSF during treatment with nusinersen, with the most substantial changes in miR-7-5p, miR-15a-5p, miR-15b-3p/5p, miR-126-5p, miR-128-2-5p, and miR-130a-3p. These miRNAs are involved in neurogenesis, neuronal differentiation and growth. The study also identified the mammalian target of rapamycin and the mitogen-activated protein kinase signaling pathways as the dominant pathways associated with these miRNAs [76].

Chen et al. investigated longitudinal changes in miR-34a, miR-34b, and miR-34c expression in CSF samples from type 1 SMA patients who were treated with nusinersen. The mean age at which the patients received their first dose of nusinersen was 8.6 months, which corresponded to the age at first sampling. Their study found that the expression of miR-34a, miR-34b, and miR-34c decreased in patients receiving nusinersen treatment. In addition, they found that the baseline miR-34b levels in the CSF are a significant predictor of treatment success after 1.5 years of therapy, as determined by the HINE score. Because changes in the miR-34 levels circulating in the CSF are correlated with improved motor function, they proposed that miR-34b and other members of the miR-34 family are promising biomarkers for predicting the patient response to treatment and may reflect treatment progress in real time by maintaining the neuromuscular junction motor endplates that are negatively affected in the early phase of SMA pathogenesis [87].

## 4. Discussion and Conclusions

MiRNAs identified as deregulated in SMA are involved in both neuronal and skeletal muscle processes. MyomiRs in SMA patients’ blood and CSF have recently been studied as biomarkers to evaluate patients’ response to nusinersen [74,80]. According to the data in Table 2, the most frequently deregulated myomiRs that were increased in SMA patients compared to controls and decreased after treatment with nusinersen include miR-1-3p, miR-133a-3p, miR-133b, and miR-206. Experimental elucidation of the functional role of these differentially expressed myomiRs in the SMA disease would increase their potential use as biomarkers.

The SMN protein is ubiquitously produced in both the cytoplasm and the nucleus, but it primarily determines selective vulnerability in motor neurons via processes that are currently unknown [16]. Although the fundamental cause of progressive muscle atrophy in SMA has traditionally been thought to be motor neuron degeneration, new investigations have also shown a skeletal muscle-specific pathological phenotype such as reduced mitochondrial activity and increased cell death through the SMN regulation of MYOD and the myomiRs miR-1 and miR-206 axis [88]. Abiusi et al. used a dual approach combining muscle cell cultures and biopsies to assess whether skeletal muscle is actively or passively implicated in SMA. While the abnormalities observed in vivo might have been either primary or secondary to denervation/atrophy processes, the changes observed in cultured cells suggest a primary issue caused by SMN deficiency [73].

As disease-modifying medicines are now available for SMA, it is critical to find clinically useful biomarkers that may predict how the illness will develop and how well a certain treatment will work. Since the SMN protein is thought to be involved in miRNA biosynthesis, miRNAs have been implicated in the etiology of SMA and may offer new knowledge regarding the disease [76]. MiRNAs have the ability to control many targets at the same time and may act in whole networks of genes affecting protein expression [89].

At this point, we wanted to further evaluate this notion. We used the RNA Association Interaction Database RAID v2 [90] to connect miRNA–mRNA and construct the miRNA–mRNA axis to gain more insight into the interconnection between these miRNAs and their role in regulating broader molecular mechanisms. Next, we utilized the Cytoscape bioinformatics software platform (v3.8.2., CytoScape Team) to visualize the molecular interaction networks and integrate them with the gene expression patterns (Figure 2).

Based on the bioinformatics predictions, we identified large clusters of mRNAs (Figure 2) representing genes targeted by two miRNAs. To gain further insights, we searched the Gene Ontology (GO) database for these genes, discovering more meaningful results. While the number of associated GO terms was relatively small, they were highly selective for neuroinflammation processes. The axis “hsa-miR-1-3p—hsa-miR-206” conditions genes including *G6PD*, *PGD*, and *TKT* (Figure 2), which belong to the concept of the pentose-phosphate shunt, which has been shown to play a role in neurological inflammatory processes [91]. There have been reports concerning elevated glucose-6-phosphate dehydrogenase (G6PDH), the rate-limiting enzyme of the pentose phosphate pathway (PPP), in experimental muscle disturbances. The PPP produces ribose, a substrate of RNA, and NADPH, which is a cofactor of fatty acid synthesis. The PPP also has a role in the glycolysis by-pathway. Konagaya et al. evaluated the G6PDH activity and RNA content in biopsied quadriceps muscles from patients with neuromuscular disorders (including amyotrophic lateral sclerosis and SMA) and found significant positive correlations between the G6PDH activity and the RNA content in motor neuron diseases [92]. Furthermore, Maier et al. found that many damaged muscle fibers in individuals with muscular disorders or lower motor neuron diseases exhibit increased activity of the two oxidative enzymes of PPP. Similarly, they discovered that experimental animals with impaired skeletal muscle, induced by various drugs or treatments, exhibited the same substantial increase in G6PDH and PGDH activity in muscle fibers, with a positive association between the activity of both enzymes. Other findings of their research indicate a favorable association between the activity of G6PDH and PGDH, on the one hand, and the activity of the non-oxidative enzymes of the PPP, the enzymes TA, TK (encoded by TKT), RPI, and RPE, on the other hand [93].

The axis “hsa-miR-133b—hsa-miR-133a-3p” conditions the genes *FGFR1*, *SP1*, and *TGFB1*, which belong to the concept of positive regulation of blood vessel endothelial cell migration, which could be related to SMA via microvasculopathy.

In SMA patients, Zhou et al. found an unfavorable balance between endothelial injury and repair, which was reflected in an increased number of circulating endothelial cells and reduced endothelial progenitor cell counts in the blood. Biological markers of endothelial damage in cultured endothelial cells from SMA patients were associated with disease severity and improved following SMN restoration treatment. In cultured human and mouse endothelial cells, they identified autonomous defects in angiogenesis and blood vessel formation as the primary molecular mechanism of microvascular illness caused by SMN depletion. Their results, using cellular and vascular biomarkers, highlight microvasculopathy as a major feature of SMA [94].

To conclude, in this review, we have gathered data on miRNAs implicated in SMA pathogenesis and their potential as biomarkers. We have demonstrated that comparing differences in miRNA expression between SMA patients and controls, as well as before and after treatment, opens up possibilities not only for monitoring disease progression and treatment efficacy but also for gaining insight into the broader complex molecular mechanisms and networks underlying SMA. This understanding could aid in the design of future therapeutic targets. Although much work remains, we believe these findings point in the right direction, toward discovering novel clinically meaningful biomarkers and treatment targets for SMA.

## Figures and Tables

**Figure 1 biomedicines-12-02428-f001:**
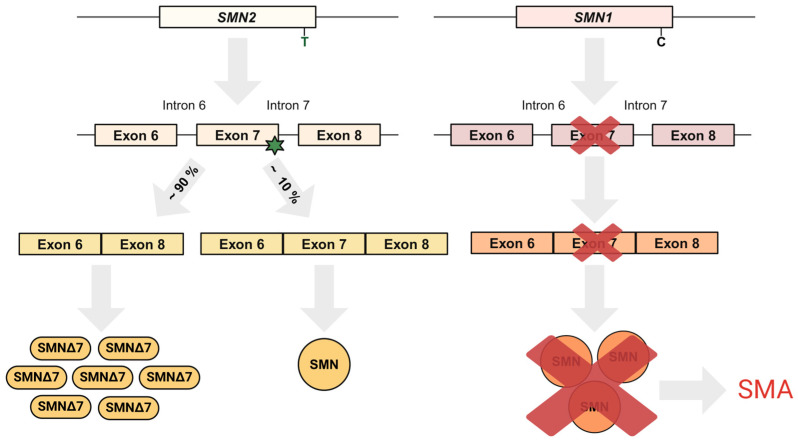
Schematic representation of *SMN1* and *SMN2* splicing. SMA is most commonly caused by loss of the *SMN1* gene, whose function is to produce a functional full-length SMN protein. However, the human *SMN* gene is duplicated, resulting in the *SMN2* gene, which differs from *SMN1* by several nucleotide changes, with the most crucial being the C-to-T transition in exon 7. This transition results in the exclusion of exon 7 in ∼90% of *SMN2* transcripts. Most *SMN2*-derived mRNAs are translated into a truncated, nonfunctional SMNΔ7 protein. The remaining 10% of *SMN2* transcripts contain exon 7 and produce the functional full-length SMN, which, however, cannot fully compensate for the loss of *SMN1*. SMA, spinal muscular atrophy; *SMN1*, survival motor neuron 1; *SMN2*, survival motor neuron 2; SMN, survival motor neuron.

**Figure 2 biomedicines-12-02428-f002:**
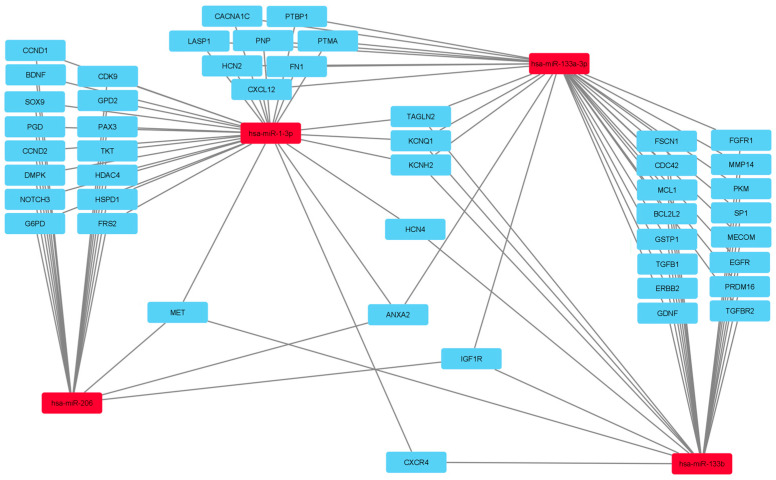
RAID v2 and Cytoscape were used to generate a projected miRNA–mRNA network implicated in SMA. The red cells represent miRNAs and the blue cells represent mRNA and “hsa” denotes Homo sapiens.

**Table 1 biomedicines-12-02428-t001:** Classification and clinical manifestation of SMA [23,25,26,27,28,29].

SMA Type	*SMN2* Copy Number	Age at Symptom Onset	Maximal Function	Natural Disease History	Median Survival Without Treatment
0	1	Prenatal or neonatal	Respiratory support	Faint fetal movements, asphyxia, contractures, severe weakness at birth	Weeks
1	2	0–6 months	Unable to sit unaided	Severe, generalized muscle weakness, hypotonia, feeding problems, respiratory failure	<2 years
2	3	<18 months	Non-ambulant patients able to sit independently	Progressive muscle weakness, scoliosis	>4 years
3a	3	<3 years	Ambulant patients with childhood-onset SMA	Proximal muscle weakness	Normal
3b	4	>3 years			
4	4–6	Second or third decade	Ambulant patients with adult-onset SMA	Mild hypotonia, gradual muscle weakening	Normal

SMA, spinal muscular atrophy; *SMN2*; survival motor neuron 2.

**Table 2 biomedicines-12-02428-t002:** miRNA biomarkers for SMA.

Biofluid	Treatment Time Points	SMA Patients (n)	Patient Age at Sampling	Controls (n)	Potential miRNA Biomarkers	Measuring Method	Results	Author, Year
Serum	/	10 (6 type 2, 4 type 3)	4–14 yrs	7 HC	miR-9, miR-132, miR-206	RT-qPCR	Increased levels of miR-9, miR-132 (SMA type 2)	Catapano et al. (2016) [78]
Serum	/	23 (17 type 2, 6 type 3)	6.86 ± 3.33 yrs	17 HC	miR-206, miR-133a, miR-133b, miR-1	RT-qPCR	Increased levels of miR-206	Malacarne et al. (2021) [79]
Serum	/	51 (3 type 1, 21 type 2, 26 type 3, 1 type 4)	17.3 ± 19.2 yrs	37 HC	miR-181a-5p, miR-324-5p, miR-451a	NGS, RT-qPCR	Increased miRNA levels	Abiusi et al. (2021) [73]
Serum	Baseline, 6 months	21 (type 2, 3)	5.18 ±3.42 yrs	/	miR-133a, miR-133b, miR-206, miR-1	RT-qPCR	Decreased myomiR levels with treatment; reduction of miR-133a as a biomarker of therapeutic effect	Bonanno et al. (2020) [80]
CSF	Baseline, various months after the treatment	12 (type 1–3)	0.2–20 yrs	/	miR-132, miR-218, miR-9, miR-23a, miR-431, miR-146a	RT-qPCR	Increased miRNA levels with treatment	Welby et al. (2022) [86]
CSF	Baseline, 6 months	34 (type 2, 3)	11.0 (1.7–56.6) yrs	/	miR-206, miR-133a-3p	NGS	Decreased miRNA baseline levels associated with better treatment response	Magen et al. (2022) [74]
Plasma	Baseline, 2 months, 6 months	10 (type 2, 3; discovery cohort) 22 (type 1; treatment cohort)	10.2 yrs (discovery cohort) 19 mo (therapeutic cohort)	7 HC	miR-107, miR-142-5p, miR-335-5p, miR-423-3p, miR-660-5p, miR-378a-3p, miR-23a-3p *	NGS, RT-qPCR	Positive correlation of the baseline * or baseline and 2-month miRNA levels with functional improvement	Zaharieva et al. (2022) [75]
CSF	Baseline, 6 months	6 infants	16–390 d	/	miR-7-5p, miR-15a-5p, miR-15b-3p/5p, miR-126-5p, miR-128-2-5p, miR-130a-3p	NGS	Decreased miRNA baseline levels	D’Silva et al. (2023) [76]
CSF	Baseline, 1.5 years	7 (type 1)	8.6 mo	/	miR-34a, miR-34b, and miR-34c	RT-qPCR	Decreased miRNA levels with treatment; baseline miR-34 levels proved predictive of patient motor skills	Chen et al. (2023) [87]

* Only baseline miR-23a levels positively correlated with functional improvement. Abbreviations: SMA, spinal muscular atrophy; CSF, cerebrospinal fluid; HC, healthy control; RT-qPCR, reverse transcription-quantitative polymerase chain reaction; NGS, next-generation sequencing.

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
