# Peer review of "MicroRNAs as Biomarkers in Spinal Muscular Atrophy"

_biomedicines, 2024, doi:10.3390/biomedicines12112428_

Round 1
Reviewer 1 Report (New Reviewer)
Comments and Suggestions for Authors
Barbo et al performed a narrative review and gathered evidence on miRNAs association with SMA pathogenesis and their potential use as circulating biomarkers. Several issues need to be address to further improve the quality of their manuscript.
Generally, in the abstract and introduction common knowledge facts for this disease are reproduced rather than highlightening the need for conducting this review, which is related with the lack of any available biomarkers to monitor SMA disease progression and therapeutic outcome. Thus, authors should spend at least one paragraph elaborating on this need for biomarkers for SMA which is the reason why it is important that miRNA has been shown promising results. Authors should counsel and cite similar narrative (PMID: 37289324) and systematic reviews (PMID: 34484951, PMID: 35861914, PMID: 35115230) and declare in which ways their review differs and is needed.
Abstract
Authors should trim general info on SMA , state why their review is needed along with their main findings.
“We also demonstrated that differences in miRNA expression between patients and controls, as well as between patients before and after treatment, opens up possibilities for not only monitoring disease progression and treatment, but also for gaining insight into the disease's broader complex molecular mechanisms and networks, which may aid in the design of future therapeutic targets.” This sentence is a bit confusing. How did the authors demonstrated this through conducting a review? Authors should replace the phrase “We also demonstrated that” with “based on current evidence the”
Paragraph 1.3. Classification and Clinical manifestation: this should be trimmed, as this is not the goal of this review. Authors should just keep 2-3 sentences and table 1. Instead a paragraph on SMA biomarkers should be added. Authors could focus on extraneuronal SMA manifestations that may be driven by SMN protein involvement in miRNA biosynthesis (PMID: 32409122).
Paragraph 1.4: Authors should revise this paragraph and be more careful with the citations provided:
“There are currently three therapies approved by the 127 U.S. Food and Drug Administration (FDA) and the European Medicines Agency (EMA), 128 which are designed to increase SMN production (46).” Remove citation 46 that represents a literature review with the respective citations of EMA and FDA approval for each drug (for example for nusinersen is citation 48,49).
“Nusinersen (Spinraza; Biogen), as the first approved SMA treatment, is an antisense oligonucleotide designed to modify the splicing of SMN2 pre-mRNA (47).” Authors should explain why they have chosen this particular citation and should better use the citations of the relevant RCTs (PMID: 29443664, PMID: 29091570) and real world evidence of nusinersen effectiveness in adults (PMID: 32199097, PMID: 35178673)
Paragraph 2:
Some info on the techniques of miRNA measurement should be provided.
“With the development of new therapies for SMA come challenges in finding reliable measures for evaluating their effectiveness (68, 69). “ Authors should replace these citations off narrative reviews with real world studies or with SR & meta-analyses if available.
“Circulating miRNAs are currently being used as novel clinical indicators for predict-170 ing the outcome of several diseases.” Please elaborate with examples.
2.1.1. Circulating miRNA as potential diagnostic biomarkers for SMA: Please define the age of each study population. Is there any difference between adults and children?
“Despite the advancement of innovative molecular and gene therapies for SMA, nusinersen remains the most frequently used disease-modifying SMA treatment.” This statement is based on which evidence?
At Table 2 info on patients age must be presented
3. Discussion and Conclusion
Please reorganize the discussion as it is a bit chaotic. The first paragraph should focus on summarizing the main points of the review . For example “MiRNAs identified as deregulated in SMA are involved in both neuronal and skeletal muscle processes. MyomiRs in SMA patients' blood and CSF have recently been studied as biomarkers to evaluate patients' response to nusinersen (76, 84). According to the data in Table 2, the most frequently deregulated myomiRs that were increased in SMA patients compared to controls and decreased after treatment with nusinersen include hsa-miR-1-317 3p, hsa-miR-133a-3p, hsa-miR-133b, hsa-miR-206. Experimental elucidation of the functional role of these differentially expressed myomiRs in the SMA disease would even increase their potential use as biomarkers.”
The purpose of the review should be moved at the last paragraph of the introduction rather that been stated at the discussion.
“We have also demonstrated that 322 comparing differences in miRNA expression between patients and controls, and patients 323 after treatment, opens up possibilities not only for monitoring disease progression and 324 treatment, but also for gaining insight into the disease's broader complex molecular 325 mechanisms and networks, which may aid in the design of future therapeutic targets.” Same sentence with abstract. How did authors achieve that?
“Although much more work remains to be done, we believe these findings point us in 331 the right direction in our quest to discover novel clinically meaningful biomarkers and 332 treatment targets for SMA.” Which findings? This was supposed to be a review.
Comments on the Quality of English LanguageMinor editing of English language required. Reorganization of the discussion must be made.
Author Response
All lines in the revised manuscript are referred to document with visible »Track Changes«
Comment 1
Barbo et al performed a narrative review and gathered evidence on miRNAs association with SMA pathogenesis and their potential use as circulating biomarkers. Several issues need to be address to further improve the quality of their manuscript.
Generally, in the abstract and introduction common knowledge facts for this disease are reproduced rather than highlightening the need for conducting this review, which is related with the lack of any available biomarkers to monitor SMA disease progression and therapeutic outcome. Thus, authors should spend at least one paragraph elaborating on this need for biomarkers for SMA which is the reason why it is important that miRNA has been shown promising results. Authors should counsel and cite similar narrative (PMID: 37289324) and systematic reviews (PMID: 34484951, PMID: 35861914, PMID: 35115230) and declare in which ways their review differs and is needed.
Response 1
Thank you for your comment.
We elaborated the need for biomarkers for SMA on several places in the revised text:
A new paragraph was added (1.5 SMA biomarkers; lines: 178-239 in the revised version); and in lines 284-292
Comment 2
Abstract
Authors should trim general info on SMA, state why their review is needed along with their main findings.
“We also demonstrated that differences in miRNA expression between patients and controls, as well as between patients before and after treatment, opens up possibilities for not only monitoring disease progression and treatment, but also for gaining insight into the disease's broader complex molecular mechanisms and networks, which may aid in the design of future therapeutic targets.” This sentence is a bit confusing. How did the authors demonstrated this through conducting a review? Authors should replace the phrase “We also demonstrated that” with “based on current evidence the”
Response 2
Thank you for your comment.
We rewrote the abstract (lines: 12-27 in the revised version).
Comment 3
Paragraph 1.3. Classification and Clinical manifestation: this should be trimmed, as this is not the goal of this review. Authors should just keep 2-3 sentences and table 1. Instead a paragraph on SMA biomarkers should be added. Authors could focus on extraneuronal SMA manifestations that may be driven by SMN protein involvement in miRNA biosynthesis (PMID: 32409122).
Response 3
Thank you for your comment.
We trimmed the text on clinical manifestation and the paragraph on SMA biomarkers was added: 1.5 SMA biomarkers; lines: 178-239 in the revised version
Comment 3
Paragraph 1.4: Authors should revise this paragraph and be more careful with the citations provided:
“There are currently three therapies approved by the 127 U.S. Food and Drug Administration (FDA) and the European Medicines Agency (EMA), 128 which are designed to increase SMN production (46).” Remove citation 46 that represents a literature review with the respective citations of EMA and FDA approval for each drug (for example for nusinersen is citation 48,49).
“Nusinersen (Spinraza; Biogen), as the first approved SMA treatment, is an antisense oligonucleotide designed to modify the splicing of SMN2 pre-mRNA (47).” Authors should explain why they have chosen this particular citation and should better use the citations of the relevant RCTs (PMID: 29443664, PMID: 29091570) and real world evidence of nusinersen effectiveness in adults (PMID: 32199097, PMID: 35178673)
Response 3
Thank you for your comment.
Citation has been removed and new citations of EMA and FDA approval for each drug were added.
Suggested references (PMID: 29443664, PMID: 29091570, PMID: 32199097, PMID: 35178673) were used instead.
Paragraph 1.4 Disease Management lines: 154-177
Comment 4
Paragraph 2:
Some info on the techniques of miRNA measurement should be provided.
Response 4
Thank you for your comment.
We added the following to the Paragraph 2: Micro RNAs in SMA; lines 252-268
The RT-qPCR method is a gold standard for the detection of miRNAs. It offers high sensitivity and single nucleotide specificity while being cost-effective at the same time, however, enables only relative quantification (56-58). The method involves two steps: RT to synthesize cDNA from the miRNA target, followed by PCR amplification monitored by fluorescence. However, the adaptation of RT-qPCR for miRNA quanti-fication requires special primers, such as stem-loop primers (57) or two-tailed primers (59), as well as optimizations such as poly(A) strategies (56) or ligation-based PCR methods (60, 61).
Another technique for miRNA detection is next generation sequencing (NGS), a powerful tool for miRNA profiling. It enables the discovery of numerous miRNAs by sequencing millions to billions of RNA sequences within a short time frame (62). The method includes RNA extraction, adapter ligation, RT, PCR amplification and se-quencing. One of its major advantages is its high multiplexing capability, which ena-bles the detection of all RNAs in a sample without specific probes or primers (63). However, NGS has some limitations, including the need for pre-amplification, poten-tial sequence dependent biases, lower sensitivity for rare miRNAs and high cost. Nev-ertheless, it is emerging as a benchmark for identifying disease-related miRNA signa-ture, as shown in the following studies in the field of SMA (64-67)
Comment 5
“With the development of new therapies for SMA come challenges in finding reliable measures for evaluating their effectiveness (68, 69).“ Authors should replace these citations off narrative reviews with real world studies or with SR & meta-analyses if available.
Response 5
Thank you for your comment.
We removed this sentence becuse we think it does not contribute essentially to the review.
Comment 6
“Circulating miRNAs are currently being used as novel clinical indicators for predict-170 ing the outcome of several diseases.” Please elaborate with examples.
Response 6
Thank you for your comment.
We corrected the sentence and provided the examples: lines: 300-301
Circulating miRNAs are currently emerging as novel clinical indicators for pre-dicting the outcome of several diseases (Basak 2016; Carter 2017, Eyileten 2018).
- Basak I, Patil KS, Alves G, Larsen JP, Møller SG. microRNAs as neuroregulators, biomarkers and therapeutic agents in neurodegenerative diseases. Cell Mol Life Sci. 2016;73(4):811-27.
- Carter JV, Galbraith NJ, Yang D, Burton JF, Walker SP, Galandiuk S. Blood-based microRNAs as biomarkers for the diagnosis of colorectal cancer: a systematic review and meta-analysis. Br J Cancer. 2017;116(6):762-74.
- Eyileten C, Wicik Z, De Rosa S, Mirowska-Guzel D, Soplinska A, Indolfi C, et al. MicroRNAs as Diagnostic and Prognostic Biomarkers in Ischemic Stroke-A Comprehensive Review and Bioinformatic Analysis. Cells. 2018;7(12).
Comment 7
2.1.1. Circulating miRNA as potential diagnostic biomarkers for SMA: Please define the age of each study population. Is there any difference between adults and children?
Response 7
Thank you for your comment.
There is not enough data on adult SMA patients to answer this question. The ages for each study population were added:
- Catapano et al. evaluated three selected miRNAs – miR-9, miR-206, and miR-132 – in type 2 and type 3 SMA patients aged 4 to 14 years.
- Malacarne et al. investigated the expression of selected muscle-specific miRNAs (myomiRs) – miR-206, miR-133a, miR-133b, and miR-1 – in the serum of pediatric SMA patients, aged 6.86 ± 3.33 years, to establish the role of myomiR as a noninvasive biomarker.
- Abiusi and co-workers first analyzed the miRNome of muscle samples from SMA patients (median age 1.8 years) and compared it with controls, followed by evaluation of over 100 miRNAs which were found to be differentially expressed in muscle sam-ples. They identified 24 differentially expressed miRNAs and validated them in a larg-er cohort of 51 SMA patients (mean 17.3 ± 19.2 years).
- A study by Bonanno et al. looked at using certain serum-derived myomiRs, name-ly miR-133a, miR-133b, miR-1, and miR-206, as biomarkers to monitor the effects of nusinersen therapy in pediatric SMA type 2 and 3 patients, with a mean age of 5.18 ±3.42 years.
- The study included a total of 12 SMA patients, ranging in age from 2 months to 20 years, with SMA type 1 through type 3 SMA (Welby et al. 2022).
- Their study included 34 individuals with late-onset SMA, with a median age of 11.0 year (range: 1.7–56.6 years). (Magen et al. 2022).
- In a similar study by Zaharieva et al. (76), NGS-based miRNA profiling identified 69 miRNAs that were significantly dysregulated in SMA type 2 and type 3 patients, aged 7 to 16.1 years, compared to healthy controls.
- In the exploratory study by D'Silva et al., miRNA sequencing was used to examine the differential expression of miRNAs in the CSF of six children, aged 16 to 390 days at the collection of first CSF specimen, who were treated with nusinersen.
- The mean age at which the patients received their first dose of nusinersen was 6 months, which corresponded to the age at first sampling. (Chen et al. 2023).
Comment 8
“Despite the advancement of innovative molecular and gene therapies for SMA, nusinersen remains the most frequently used disease-modifying SMA treatment.” This statement is based on which evidence?
Response 8
Thank you for your comment.
We corrected the sentence and provided the examples: lines: 341-343
Despite the advancement of innovative molecular and gene therapies for SMA, nusinersen continues to be the most extensively studied disease-modifying SMA treatment (Giess et al., 2024).
Comment 9
At Table 2 info on patients age must be presented
Response 9
Thank you for your comment.
New column with ages was added
Comment 10
- Discussion and Conclusion
Please reorganize the discussion as it is a bit chaotic. The first paragraph should focus on summarizing the main points of the review . For example “MiRNAs identified as deregulated in SMA are involved in both neuronal and skeletal muscle processes. MyomiRs in SMA patients' blood and CSF have recently been studied as biomarkers to evaluate patients' response to nusinersen (76, 84). According to the data in Table 2, the most frequently deregulated myomiRs that were increased in SMA patients compared to controls and decreased after treatment with nusinersen include hsa-miR-1-317 3p, hsa-miR-133a-3p, hsa-miR-133b, hsa-miR-206. Experimental elucidation of the functional role of these differentially expressed myomiRs in the SMA disease would even increase their potential use as biomarkers.”
The purpose of the review should be moved at the last paragraph of the introduction rather that been stated at the discussion.
“We have also demonstrated that 322 comparing differences in miRNA expression between patients and controls, and patients 323 after treatment, opens up possibilities not only for monitoring disease progression and 324 treatment, but also for gaining insight into the disease's broader complex molecular 325 mechanisms and networks, which may aid in the design of future therapeutic targets.” Same sentence with abstract. How did authors achieve that?
“Although much more work remains to be done, we believe these findings point us in 331 the right direction in our quest to discover novel clinically meaningful biomarkers and 332 treatment targets for SMA.” Which findings? This was supposed to be a review.
Minor editing of English language required. Reorganization of the discussion must be made.
Response 10
Thank you for your comment.
We rewrote the whole 3. Discussion and Conclusion
Lines 442-552
Reviewer 2 Report (New Reviewer)
Comments and Suggestions for Authors
The review presented by Maruša Barbo and co-authors is devoted to application of miRNAs as potential biomarkers in spinal muscular atrophy. The authors reviewed most of the publications relevant to the field and the manuscript is scientifically sound and well-written. Nonetheless, a few major and minor concerns need to be addressed prior to publication.
MAJOR
1. It is highly recommended to add more information on role of dysregulation or abnormal expression of specific miRNAs in spinal muscular atrophy.
2. Additionally, it would be beneficial to include a figure illustrating the most relevant miRNAs used as biomarkers for SMA, their targets, and their links to the pathology of SMA.
MINOR
Line 42: The information needs to be updated. Now it is known that the genes are differed by 14 nucleotide changes and one insertion (Butchbach, M.E.R. Genomic Variability in the Survival Motor Neuron Genes (SMN1 and SMN2): Implications for Spinal Muscular Atrophy Phenotype and Therapeutics Development. Int. J. Mol. Sci. 2021, 22, 7896, doi:10.3390/ijms22157896).
Line 46: “ranges from 0 to 5” is more correctly. This fact needs to be clarified. Please, explain cases when no SMN2 gene is present in SMA patients. Actually, absence of SMN1 and SMN2 leads to prenatal death in animal models.
Figure 1: Please, correct grammatical error – replace exon instead of ekson.
Line 59: “growth, and survival” – no comma needed
Line 65: “SMN2 gene, which differs from SMN1 by a single nucleotide change” - inaccurate formulation, cause genes differs in more nucleotides, but this one is the most significant
Line 145: “studies on the causes of SMA are now focusing on RNA metabolism” – the formulation also seems not quite accurate, since the cause of SMA is mutations within the SMN1 gene.
Line 151: “function, and survival” – no comma needed
Line 190: “myomiR” occurs in text for the first time, while the definition occurs in line 217
Line 317: “hsa-miR” – give definition
Author Response
All lines in the revised manuscript are referred to document with visible »Track Changes«
Comment 1
The review presented by Maruša Barbo and co-authors is devoted to application of miRNAs as potential biomarkers in spinal muscular atrophy. The authors reviewed most of the publications relevant to the field and the manuscript is scientifically sound and well-written. Nonetheless, a few major and minor concerns need to be addressed prior to publication.
MAJOR
- It is highly recommended to add more information on role of dysregulation or abnormal expression of specific miRNAs in spinal muscular atrophy.
- Additionally, it would be beneficial to include a figure illustrating the most relevant miRNAs used as biomarkers for SMA, their targets, and their links to the pathology of SMA.
Response 1
Thank you for your comment.
We added the Figure 2 and corresponding explanation in the 3. Discussion and Conclusion
Lines 442-552
Comment 2
MINOR
Line 42: The information needs to be updated. Now it is known that the genes are differed by 14 nucleotide changes and one insertion (Butchbach, M.E.R. Genomic Variability in the Survival Motor Neuron Genes (SMN1 and SMN2): Implications for Spinal Muscular Atrophy Phenotype and Therapeutics Development. Int. J. Mol. Sci. 2021, 22, 7896, doi:10.3390/ijms22157896).
Response 2
Thank you for your comment.
We updated the information:
SMN1 and SMN2 genes differ by 14 nucleotides and one insertion, the most important nucleotide change is a C-to-T transition located in the coding region, in exon 7 (Butchbach, M.E.R. Genomic Variability in the Survival Motor Neuron Genes (SMN1 and SMN2): Implications for Spinal Muscular Atrophy Phenotype and Therapeutics Development. Int. J. Mol. Sci. 2021, 22, 7896, doi:10.3390/ijms22157896).
See Lines 58-60
Comment 3
Line 46: “ranges from 0 to 5” is more correctly. This fact needs to be clarified. Please, explain cases when no SMN2 gene is present in SMA patients. Actually, absence of SMN1 and SMN2 leads to prenatal death in animal models.
Response 3
Thank you for your comment.
We added this sentence:
A lower number of SMN2 copies is associated with lower levels of full-length SMN protein, while the complete absence of SMN protein is lethal (Burghes AH. When is a deletion not a deletion? When it is converted. Am J Hum Genet. 1997 Jul;61(1):9-15. doi: 10.1086/513913. PMID: 9245977; PMCID: PMC1715883). Therefore, the severity of SMA varies, at least partially, depending on the number of SMN2 copies that patients carry
Lines 63-65
Comment 4
Figure 1: Please, correct grammatical error – replace exon instead of ekson.
Response 4
Thank you for your comment.
We corrected in the Figure 1
Comment 5
Line 59: “growth, and survival” – no comma needed
Response 5
Thank you for your comment.
We corrected
Comment 6
Line 65: “SMN2 gene, which differs from SMN1 by a single nucleotide change” - inaccurate formulation, cause genes differs in more nucleotides, but this one is the most significant
Response 6
Thank you for your comment.
We corrected
However, the human SMN gene is duplicated, resulting in the SMN2 gene, which differs from SMN1 by several nucleotide changes, with the most crucial being the C-to-T transition in exon 7.
Lines: 93-94
Comment 7
Line 145: “studies on the causes of SMA are now focusing on RNA metabolism” – the formulation also seems not quite accurate, since the cause of SMA is mutations within the SMN1 gene.
Response 7
Thank you for your comment.
The sentence was corrected:
Researchers have discovered that many SMN-associated modifiers play a role in both coding and non-coding RNA (ncRNA) processing, so studies of the molecular mechanisms underlying SMA are now focusing on RNA metabolism.
Lines 242-244
Comment 8
Line 151: “function, and survival” – no comma needed
Response 8
Thank you for your comment.
We corrected
Comment 9
Line 190: “myomiR” occurs in text for the first time, while the definition occurs in line 217
Response 9
Thank you for your comment.
We corrected
Line 321
Comment 10
Line 317: “hsa-miR” – give definition
Response 10
Thank you for your comment.
We defined hsa- in the description under the Figure 2.
Round 2
Reviewer 2 Report (New Reviewer)
Comments and Suggestions for Authors
Please, see attached file.

Round 3
Reviewer 2 Report (New Reviewer)
Comments and Suggestions for Authors
The authors successfully improved the manuscript.
This manuscript is a resubmission of an earlier submission. The following is a list of the peer review reports and author responses from that submission.
Round 1
Reviewer 1 Report
Comments and Suggestions for Authors
The authors have described the potential for microRNAs as biomarkers for the pediatric-onset motor neuron disease spinal muscular atrophy. While this manuscript is labeled a review article, it actually reads as a hypothesis or perspective piece.
The genetics of spinal muscular atrophy were presented in a confused manner and some of the information presented is not entirely correct or current.
The authors also present new data in this review in the form of a novel network of mRNA-microRNA-lncRNA in SMA. The authors should have presented this novel analysis as a primary research papers, with the appropriate controls and validation.
Reviewer 2 Report
Comments and Suggestions for Authors
The authors have summarized the miRNA associated with SMA and evaluate their potential as a biomarker for SMA. The manuscript is well written and properly organised. The references are upto date. I have only some minor comments
1. Please add a figure describing the splicing of SMN1 and SMN2 ( Line 39).
2. Please add a table for summarizing section - Classification and Clinical manifestation.
3. Please make a separate section for circular miRNA.
Comments on the Quality of English LanguageThere are minor spelling mistakes in the text which needs to be corrected.
Reviewer 3 Report
Comments and Suggestions for Authors
In this review article entitled “MicroRNAs as biomarkers in spinal muscular atrophy”, the authors summarized the recent advances in the study of microRNA in spinal muscular atrophy (SMA).
The SMN1 gene is ubiquitously expressed as a housekeeping gene, and its functional product, the SMN protein, plays a critical role in cell homeostasis and takes part in a variety of cellular mechanisms. However, we do not fully understand how the SMN protein works with other genes and/or proteins. To fully elucidate the machinery of the SMN protein, many studies on the partner proteins bound to the SMN protein have been reported. Recently, researchers have payed attention to microRNA synthesis in which the SMN protein involves. Many kinds of microRNA associated with the SMN protein may be modifiers of cellular functions in SMA patients, or they may be at least biomarkers of SMA phenotype.
The authors have done a good job summarizing recent miRNA research to allow the reader to quickly understand the progress of research in this field. Readers will also appreciate this review article for a quick overview of SMA.
This paper is clearly written, and I learned a lot from this paper, and would like to thank the authors.
Although this paper is almost perfect, I found a very small flaw in the manuscript.
[VERY SMALL FLAW]
Line 129: NDs; The authors should also indicate the full form of this word.
Round 2
Reviewer 1 Report
Comments and Suggestions for Authors
This manuscript still reads as a hypothesis paper as opposed to a review of the current literature. Additionally, the description of the genetics of SMA--while improved--is still not entirely accurate and cites other review articles instead of the primary literature (for example, the description of the SMN2c.859G>C cites a review article instead of the primary paper).
The figure describing the differences between SMN1 and SMN2 is not entirely accurate and misleading. According to this figure, a proportion of SMN2 mRNAs are converted into SMN1, which is definitely not the case.
As a more formal point, SMA and ALS are generally considered to be motor neuron diseases as opposed to neuromuscular disorders.